

# Analysis of disaster characteristics and emergency response of the Jiuzhaigou earthquake

WANG Wei[1], CHEN Hong[2**], XU Aihui[2], QU Minhao[3]

[1]Institute of Engineering Mechanics, China Earthquake Administration, Harbin 150080, China
[2]Institute of Crustal Dynamics, China Earthquake Administration, Beijing 100085, China
[3]National Earthquake Response Support Service, Beijing 100049, China

*Correspondence to*: Chen Hong (chenhongicd@163.com)

**Abstract.** This paper provides a detailed introduction to the disaster situation of the August 8, 2017, Ms 7.0 earthquake that occurred in Jiuzhaigou County, Sichuan Province, China, and specifically describes the emergency response activities of all

levels of the government, various departments, rescue teams, enterprises and public institutions as well as social organizations. The characteristics of the earthquake disaster and the emergency responses are analysed and summarized. The emergency response activities during the Jiuzhaigou earthquake exhibited three distinct characteristics relative to those during similar earthquakes. The response level and resource mobilization were appropriately and sensibly adjusted according to the development of the disaster, and various departments worked closely together to conduct multi-sector joint rescue efforts.

Moreover, professional rescue forces and participating social organizations were more rationally mobilized. A set of effective disaster relief command and coordination mechanisms were established for cooperation between multiple departments and the participation of many social organizations under the leadership of the local government. Finally, newer and more effective technologies played an important role in the emergency response and rescue efforts following this earthquake.

**Key words**: Jiuzhaigou earthquake; earthquake disaster; emergency response; emergency rescue

## 1 Introduction

China is geographically positioned between the circum-Pacific seismic belt and the Eurasian seismic zone, and thus, most of the earthquakes occur on the mainland. China frequently experiences severe, high-intensity, shallow earthquakes with a wide

distribution. Multiple public health crises have affected China since 2003, including the SARS (severe acute respiratory syndrome) and bird flu epidemics, which highlight the importance of emergency response teams and activities. In January 2006, the State Council promulgated the "National Contingency Plan for Public Emergencies"(General Office of the State Council, 2006). This general contingency plan divides emergencies into four categories: natural disasters, accidental disasters, public health events and social security events. According to the degree of damage in addition to the urgency and development

of the public emergency, the response is divided into four levels: grade I (especially significant), grade II (major), grade III



(larger) and grade IV (general). Simultaneously, 25 special plans and 80 department plans have been released based on the general contingency plan (Liu and Li, 2010). The special plans include five types of natural disaster emergency plan; among these plans, the National Earthquake Contingency Plan incorporates emergency response activities for earthquake disasters (including volcanic disasters)(General Office of the State Council, 2012). Based on the Emergency Response Law of the

People's Republic of China(The 10th Standing Committee of the National People's Congress, 2007), the Law of the People's Republic of China on Protecting Against and Mitigating Earthquake Disasters(Standing Committee of the National People's Congress, 2008), and the National Earthquake Contingency Plan, the China Seismological Bureau formulated the Earthquake Contingency Plan of the China Seismological Bureau(China Seismological Bureau, 2013). All levels of the government of the People's Republic of China, relevant departments, departments in charge of earthquake-related work, units directly under the

China Earthquake Administration, key enterprises and institutions, schools, hospitals, and personnel-intensive locations, have established their own earthquake contingency plans.

The response mechanism of China Earthquake Administration has undergone constant improvement through practical testing and summary reviews of the magnitude 8 Wenchuan earthquake of 2008, the magnitude 7.1 Yushu earthquake of 2010, and the magnitude 7 Lushan earthquake of 2013. Unfortunately, the local government and social forces were not fully mobilized

during the response to the Wenchuan earthquake. After the Wenchuan earthquake, the Chinese government revised the Law of the People's Republic of China on Protecting Against and Mitigating Earthquake Disasters, strengthened the functions of all levels of government, and enhanced departmental responsibilities and social participation. Moreover, following the Wenchuan earthquake, the State Council formed a first-level emergency response scenario and established the National Earthquake Relief Headquarters with the Prime Minister as the Supreme Commander. Rescue resources under the direct

command of the State Council are deployed to disaster regions to conduct rescue operations. However, some problems still exist: different rescue response resources operate in their own way without unified command and coordination, rescue resources are unevenly distributed, and social organizations and volunteer teams lack an overall management structure.

The local government played a major role in the response and rescue operations following the Yushu earthquake in 2011 by establishing earthquake relief headquarters directed by the local governments of the disaster-stricken areas to take full charge

of the rescue operations. Although the State Council had previously established the National Earthquake Relief Headquarters, its main purpose is to organize and coordinate the rescue operations of central departments and units mainly in accordance with the specific needs of disaster areas. The localities were fully mobilized during their response to the Yushu earthquake, and consequently, the rescue operations were more ordered. However, due to the geographical environment and limited traffic conditions in the disaster areas, rescue teams were unable to adapt to the hypoxia that they encountered atop the plateau, and

some other teams were unable to efficiently travel to the disaster areas due to road congestion (Wang, 2010). Therefore, after the Yushu earthquake, China revised the National Earthquake Contingency Plan to note that a first-level emergency response scenario should be activated when dealing with especially significant earthquake disasters and that emergency and response operations in a disaster area should be led by the provincial earthquake relief headquarters, while the State Council Earthquake Relief Headquarters should be responsible for the unified leadership, command and coordination of such earthquake response



operations nationwide. Meanwhile, a second-level response scenario should be enacted when dealing with a major earthquake, and the emergency and response operations in the disaster area should be led by the provincial earthquake relief headquarters, while the State Council Earthquake Relief Headquarters should organize and coordinate the relevant departments and units to conduct emergency efforts at the national level in accordance with the disaster conditions.

China's earthquake emergency response system had been gradually maturing when the Lushan earthquake struck. Consequently, an emergency response plan was immediately launched according to the response level, and the localities and armed forces conducted a multi-sector joint rescue. However, experience was still lacking in the management and coordination between the social organizations and volunteers, resulting in serious road congestion that professional rescue forces could not access, and therefore, the wounded were unable to be transferred.

After the Jiuzhaigou earthquake, the graded response activities of the various governmental departments at all levels were outstanding. The central governmental departments were able to efficiently activate their own emergency plans and harness their respective strengths for rescue operations. Various departments were able to work in tandem to conduct multi-sector joint rescue efforts more sensibly, that is, without the excessive dispatch of rescue forces. A command and coordination mechanism for disaster relief related to a number of departments and social organizations that was led by the local government was

established immediately after the earthquake. This new structure was capable of efficiently repairing communications, electric power and road traffic conditions in disaster areas, and airport capabilities and road traffic access to disaster areas were capable of being controlled for the first time, thereby ensuring that disaster relief materials and rescue teams arrived within disaster areas to conduct timely rescue operations.

## 2 Disaster situation

A magnitude 6.7 (Mw 6.5) earthquake struck Jiuzhaigou County, Aba Tibetan and Qiang Autonomous Prefecture, Sichuan Province, China, at 21:19:46 on August 8, 2017 (13:19:46 on August 8, 2017, GMT). The epicentre was located at 103.82 degrees east longitude and 33.20 degrees north latitude with a focal depth of 20 km. The earthquake occurred along strike-slip fault at the intersection of the Minjiang, Tazang and Huya faults. The epicentre was 39 km away from Jiuzhaigou County and 285 km away from Chengdu, the capital of Sichuan Province. The population density of Jiuzhaigou County is 15 people/km$^2$;

however, as August is a popular tourism season in Jiuzhaigou, there were approximately 38799 tourists at the nationally famous Jiuzhaigou scenic resort area when the earthquake struck month. A total of 25 people were killed in the earthquake (more than half of them died from landslides and rolling stones), 525 were injured, 6 were reported missing, and 176492 were affected, and 73671 houses were damaged to varying degrees (76 of which collapsed)(Bai, 2017). The earthquake generally presented the following disaster characteristics.

(1)    A low proportion of buildings completely collapsed. The earthquake struck a total of 8 counties in Sichuan Province and Gansu Province and exhibited a maximum intensity of 9 degrees in the epicentral area. China adopts a 12-degree intensity scale, where a degree of 9 indicates that most wood structure house and half timber house are either destroyed or seriously



damaged, few brick masonry are destroyed or most are either seriously or moderately damaged, and few frame structures are destroyed or most suffer either moderate or minor damage. The area with an intensity of 9 degrees spanned 139 km$^2$ in Jiuzhaigou County, while the total area with an intensity of 6 degrees exceeded 18295 km$^2$, as shown in Figure 1(China Seismological Bureau, 2017). The earthquake region had been previously affected by the Wenchuan earthquake in 2008, and thus, many houses in the disaster area had been rebuilt. Consequently, the buildings were constructed with a seismic fortification level of 8 degrees, and thus, the buildings in Jiuzhaigou possessed high levels of seismic fortification and favourable shock resistance. The earthquake caused 73671 houses to be damaged to varying degrees; of those, only 76 completely collapsed, accounting for only 0.1%.

(2)        Secondary disasters after the earthquake were very serious. The earthquake area is a high-elevation mountain gorge area, and Jiuzhaigou is situated in the transition zone between the Qinghai Tibet Plateau and the Sichuan Basin within a valley that is over 50 km deep. The residents and infrastructure are distributed throughout the river valley, and thus, residents and vehicles passing adjacent to the high, steep mountains are vulnerable to secondary disasters such as landslides, landslides and mudslides. Numerous landslides were caused by the earthquake, leading to the damage of road traffic, the smashing of passing vehicles and the falling of large boulders that struck more than half of the 25 victims. Figure 2 was taken by the author at the scene of the disaster the following day. The roads were narrow due to continued landslide activity, and the falling of boulders was particularly serious. Due to the numerous landslides, the natural landscape and ecological environment in the scenic areas of Jiuzhaigou were seriously damaged. Figure 3 shows remote sensing images acquired by UAVs (unmanned aerial vehicles) over the scenic areas in Jiuzhaigou, where the regions delineated with yellow lines represent landslide masses. Figure 4 shows a pair of high-quality satellite remote sensing images of Huanglong Airport at the epicentre before and after the earthquake. Six landslides were triggered following the earthquake that caused severe traffic disruptions.

(3)        Various facets of the infrastructure, such as the electricity, communications and road traffic systems, were disrupted. Jiuzhaigou experienced power failures and mobile phone signal interruptions after the earthquake. In addition, 234 base stations accounting for 43% of the total in Jiuzhaigou were out of service in the aftermath. Twenty-nine 10-kV power lines were suspended in Jiuzhaigou County after the earthquake, resulting in the outage of power to more than 1900 households in Huanglong Township. Power was restored to Jiuzhaigou County and the scenic spots two hours after the earthquake. Basic electricity service was restored to more than 1900 residents of Huanglong Township within 48 hours following the earthquake. Many places along national highway G213 and provincial highway S301 were interrupted due to collapses of high mountain collapse and rolling stone. These roads were fully accessible 20 hours after the earthquake.

## 3 Emergency response

### 3.1 Emergency response of the major governmental departments

After the earthquake, the central government quickly launched its emergency response mechanism. President Xi Jinping and Premier Li Keqiang immediately sent out important notifications and instructions that were required to understand and verify



the disaster situations, immediately organize rescue resources, save the wounded, evacuate and settle tourists and disaster-affected people, minimize casualties, strengthen earthquake monitoring efforts and prevent secondary disasters. The State Council quickly sent a working group to the scene of the disaster area to guide and coordinate the rescue operations. The working group was led by the Deputy Director General of the China Earthquake Administration and the Vice Minister of the

Ministry of Civil Affairs, which includes the Development and Reform Commission, the Ministry of Finance, the Ministry of Land and Resources, the Ministry of Housing and Urban Construction, the Ministry of Transport, and the National Health and Family Planning Commission(Cui, 2017).

Various departments of the Chinese government also launched emergency responses rapidly and conducted multi-sector joint rescue efforts that provided scientific insights and contributed to efficient earthquake relief (Table 1).

**3.2 Emergency response of the local government**

Responsible personnel from the Sichuan Provincial Party Committee and the provincial government immediately rushed to the field for rescue work. Sichuan Province and Aba Prefecture initiated a grade I earthquake emergency response(General Office of the Sichuan Provincial Government, 2012 and General Office of the Aba Prefecture Government, 2012). The Sichuan Provincial Party Committee and provincial government established an "8.8" Jiuzhaigou earthquake relief emergency

headquarters with five divisions: a medical support group, traffic management support group, telecommunication and power support group, relief material group, and publicizing and reporting group. At 06:00 on August 9, 2017 (9 hours after the earthquake), the headquarters convened their first conference to proclaim that search and rescue work and other efforts were being conducted rapidly to effectively salvage casualties, provide emergency rehabilitation to the power and telecommunications systems and ensure that the roads were clear, relocate victims and evacuate visitors, strengthen the

monitoring and early warning systems for secondary disasters, strengthen propaganda and guide public opinion, publish disaster relief information timely, and maintain the social stability of the disaster areas. In addition, the first press conference was convened in the afternoon on August 9 to announce the earthquake situation, casualties, and the progress of rescue and relief efforts to the public. The Jiuzhaigou County government established a working station for the "8.8" Earthquake Relief Social Organization and Volunteer Service Center to actively coordinate the social organizations and volunteers participating

in the earthquake relief efforts. On August 28, 2017, Sichuan Province established an "8.8" Jiuzhaigou Earthquake Post-Disaster Reconversion Committee to formulate plans for reestablishment , ecological environment modification, geological disaster prevention and control, scenic area recovery, industrial development, reestablishment of the infrastructure and public service, and reconstruction of urban and rural housing.

All of the local departments at the provincial, state, and county levels immediately initiated emergency responses and deployed

emergency rescue teams according to their superior central departments. Within 14 hours after the earthquake, 505 medical technicians and 130 ambulances had been dispatched to related counties with disaster areas. Consequently, a total of 217 wounded were treated, 166 people received ambulatory treatment and 51 people received hospitalization. The Traffic Police Divisions of Sichuan Province and Gansu Provinces dispatched more than 1700 personnel and 5325 vehicles to control traffic,



and more than 30000 people were evacuated. The Sichuan Department of Civil Affairs allocated and transported a total of 3000 cotton quilts, 3000 units of cotton clothing, 2000 tents, 1000 folding beds, 3000 sleeping bags, and 50 emergency lighting power generation assemblies to the disaster area(Department Of Civil Affairs Of Sichuan Province, 2017).

### 3.3 Emergency response of government rescue teams

When the earthquake occurred, the Western Theater Command of the People's Liberation Army dispatched land and air forces in addition to army, aviation, sapper, medical, militia, and other types of troops from the military region in Sichuan Province and the Xining joint guard centre to conduct search and rescue. By 07:40 on August 9, 2017 (10 hours after the earthquake), 4 helicopters had hurried to the disaster area from the Western Theater Command. By 18:00 on August 19 (up to 21 hours after the earthquake), the Western Theater Command had dispatched 1285 soldiers, 90 vehicles of various types, and 9 different

airplanes(Wang, 2017).

During the evening of August 8, local armed police forces in the disaster areas rapidly organized the evacuation of the on-site masses. At 16:00 on August 9, 2017 (up to 19 hours after the earthquake), the armed police had dispatched a total of 1958 soldiers and 105 units of equipment and machinery to conduct emergency rescue missions in the Jiuzhaigou earthquake-affected disaster areas; they evacuated a total of 6000 people, transferred 100 wounded, established 10 tents and excavated the

15 remains of 1 victim(Qian, 2017).

By 00:30 on August 9 (up to 3 hours after the earthquake), the main Public Security Fire Corps of Sichuan Province had dispatched all of the 21 detachments from the entire province, including 1108 soldiers, 396 vehicles, 55 life detection instruments, 30 search and rescue dogs, 33 sets of floodlights and 24 electric generators, to the field. By August 10, a total of 9 trapped victims had been rescued and more than 4300 masses had been transferred; 21 troop crawlers and passenger buses

transferred more than 1200 stranded travellers and employees in the Jiuzhaigou scenic areas.

The national earthquake disaster search and rescue teams had finished organizing by 13:00 on August 9 (16 hours after the earthquake); the teams consisted of 80 personnel from the China Seismological Bureau, a brigade in the 82[nd] army group land force of the Central Theater Command, and armed police from the General Hospital carrying 5 search and rescue dogs, 2 rescue equipment vehicles, and a total of 378 units of rescue equipment, including 8 types of life detection instruments, shoring

equipment, and break-in tools(CISAR, 2017). The national earthquake disaster search and rescue team arrived at the disaster area at 19:25 and divided into 4 squads to conduct searches for missing people in 4 villages, assess the safety of structures in the disaster areas, provide medical attention to the wounded, and assist victims in the transferring of goods and materials. In addition, the remains of a victim were transferred in the afternoon of August 12.

### 3.4 Emergency response of central enterprises and public institutions

Central enterprises and public institutions also initiated their contingency plans immediately following the earthquake. At 09:00 on August 9 (up to 12 hours after the earthquake), a total of 30 central enterprises had participated in the rescue and relief efforts by dispatching rescue forces to the front line of earthquake relief work(Wang,2017). Various power,



communications, aviation, tourism, oil play, construction, traffic and railway enterprises all joined the response and rescue efforts.

The power enterprise had restored the electric power in Jiuzhaigou County and the scenic areas within 2 hours. The communications enterprise immediately opened a hot line for people seeking their lost relatives and rushed to address emergency rescue workers. The aviation enterprise safeguarded the transport of rescuers, goods and materials. The tourism enterprise quickly reached out to tourists one by one and ensured the safety of both Chinese and foreign tourists. The oil industries opened a green channel and delivered provisions. The construction enterprise checked the roads and tunnels to ensure that they were unobstructed; among them, China State Construction Engineering Corporation saved 60 trapped tourists and helped save another 260 trapped tourists, and the emergency squad from China Railway Erju Group Corporation discovered a person in distress at 03:10 on August 9. China National Building Material Group also took advantage of their products and made preparations for the deliveries of supplies consisting of special cement goods and materials and other emergency commodities. Many other central enterprises rapidly assembled personnel and supplies for field rescue efforts; for instance, Xinxing Cathay International Group assembled and transported rescuers, tents, relief vehicles, rolling kitchens, medical care cars, water purification cars, and other logistical support equipment to the disaster area. The Disaster Express Delivery service offered by China Merchants Group established a logistics company near the epicentre on August 10 and arranged for disaster preparedness goods and materials in addition to vehicle transportation. Sinopharm Group initiated their special storage allocation contingency plan and secured an adequate supply of blood products and other drugs according to the requirements of the disaster rescue efforts.

### 3.5 Emergency response of social organizations

The Red Cross Society of China initiated a grade III response when the earthquake occurred and quickly allocated 1000 household emergency kits, 2000 cotton quilts and 2000 tents. The Sichuan Red Cross Society rapidly dispatched working groups and rescue teams to the disaster areas. Less than 2 hours after the earthquake, the Chinese Red Cross Foundation started its "Angelic Journey – Jiuzhai Action" plan and dispatched working groups to the disaster areas to understand and evaluate the conditions, after which they arranged 100000 Yuan as a contingent fund in support of the emergency rescue efforts.

The local government in Jiuzhaigou County established a working station for an "8.8" Earthquake Relief Social Organization and Volunteer Service Center to uniformly manage and deploy social rescue forces to ensure that the social organizations and volunteer teams participated in rescue operations methodically and effectively based on experience from the Wenchuan earthquake. At 23:00 on August 8 (less than 2 hours after the earthquake), the first social organization rescue team had already arrived at the disaster area and immediately began to participate in the rescue efforts. Through August 12, a total of 219 social organizations and more than 2288 volunteers had registered at the working station(Gao, 2017).

Social organization rescue operations acted in accordance with the unified arrangement of the working station, and consequently, they entered and evacuated the disaster area on time. Throughout the entire day on August 9, social organization rescue teams identified hidden dangers, transferred disaster-affected people, and assisted with the evacuation of visitors; at





dusk, did provided orderly maintenance and psychological counselling for stranded visitors in temporary shelters that had been dispatched by the working station. The working station coordinated the efforts of all involved parties, constructed volunteer service sites at main vital communications lines, stations, hospitals and other crucial locations, and dispatched more than 1200 volunteers to provide volunteer service in county towns, including transporting and distributing relief materials, guiding traffic,

providing psychological comfort, relocating stranded visitors, and caring for the wounded(China Volunteer Service Federation, 2017).

Among these teams, the Blue Sky Rescue Team sent a total of 12 teams and 259 team members, performed search and rescue for victims, evacuated the masses, constructed tents, offered medical assistance, verified information, distributed relief materials, transferred victims, and generated disaster situation assessment reports. They successfully discovered 10 victims

and the remains of 5 others who had been trapped in the scenic hinterland without communication abilities for more than 40 hours. All of the teams in the Blue Sky Rescue Team evacuated from Jiuzhaigou before 20:00 on August 12 according to the Jiuzhaigou earthquake relief headquarters.

Helicopter rescue forces also participated in the Jiuzhaigou earthquake rescue operations. Due to the physical restrictions of land-sky military helicopters, the earthquake relief headquarters dispatched two airbus helicopters from Xilinfengteng General

Avation Co. Ltd. to the dead zone of the earthquake disaster area to conduct search and rescue for the trapped, sick and wounded, during which a total of 30 victims were saved.

According to incomplete statistics, at least 33 insurance companies provided emergency measures by providing all-day, full-time insurance services, opening a green channel and allowing multi-channel claim settlements. They also simplified the procedures required for claim settlements, cancelled insurance policy claim settlements, cancelled fixed-point medical

restrictions, softened identification requirements, and provided direct, quick, door-to-door compensation service flexibly to support the rapid settlements of insurance claims in the disaster area.

## 4 Analysis and summary

1)    The Jiuzhaigou Valley earthquake exhibited three characteristics. First, the number of casualties and collapsed buildings were lower than those during other earthquakes in China with similar magnitudes over the past few years. This is

mainly because the population density at the epicentre was lower and because most of the tourists had already departed the region or travelled back to their hotels when the earthquake occurred (after 21:00 local time) despite it being peak tourist season. Furthermore, the buildings in the seismic area satisfy the 8-degree seismic fortification level requirement and demonstrate high anti-seismic properties, thereby leading to lower rates of collapse and damage. Second, the Jiuzhaigou Valley scenic area, which is located in a 9-degree seismic region, is a Chinese State Natural Reserve and has been included in *The*

*World Natural Heritage List*. Earthquakes introduce greater amounts of destruction to natural landscapes and ecological environments. Third, the seismic area is located in a mountain-canyon transition region where secondary disasters such as landslides and rockfalls are serious. More than half of the 25 deaths during the earthquake were due to rockfalls or landslides.



Traffic and roads are easy to block and difficult to clear, which increases the danger in such situations and enhances the difficulty of disaster relief.

2)      The provincial government of the disaster area immediately established a provincial-level earthquake relief headquarters to conduct the earthquake relief efforts. The State Council National Earthquake Relief Headquarters initiated a grade II emergency response and subsequently coordinated and organized the departments and commissions under the central leadership to conduct emergency rescue missions according to the needs of the disaster areas. All levels of the Chinese government, large governmental sectors, enterprises and public institutions, search and rescue teams, and social organizations initiated their corresponding contingency plans according to the situation of the disaster. Pursuant to their respective duties, each of these institutions displayed professional expertise while providing earthquake relief through close interdepartmental cooperation and multi-sector combined action. Evacuating the tourists, transferring and resettling the victims, caring for the wounded, performing search and rescue for victims, collecting and distributing relief supplies, rapidly repairing the roads, and restoring the electricity and communications were all accomplished properly and timely. Furthermore, they immediately arranged for settlements of insurance claims. All of these efforts provided social stability to the disaster area and ensured that the relief efforts were orderly. After the Ms 8.0 Wenchuan earthquake in 2008, China revised *The PRC Law On Earthquake And Calamity* and *National Preparatory Plan For Earthquake Emergency*, which is responsible for the disaster response activities of the local government, departments and commissions under the central leadership to provide professional support. By learning from the experiences of the emergency response activities during the Ms 7.1 Yushu earthquake in 2010 and the Ms 7.0 Lushan earthquake in 2013, the Chinese earthquake catastrophe emergency response system has become more mature, and the response mechanisms have greatly improved.

3)      The deployment of professional rescue forces and the response of social organizations were both more rational. Based on an examination of the disaster characteristics and development, only the national search and rescue team and the professional rescue teams that were adjacent to the disaster area were mobilized following the earthquake. The Blue Sky Rescue Team deployed 259 rescuers to the front line, although it has more than 100000 volunteers at the national scale. Therefore, because rescue forces were not over-deployed, traffic jams were not formed due to the over-deployment of rescue forces.

4)      By summarizing the experiences from the Wenchuan and Yushu earthquakes, the accessibility of the roads and airports in the disaster area were rapidly managed and controlled to ensure the timely arrival of relief materials and rescue teams. In addition, an effective command and coordination structure was established for the social organizations that joined the relief efforts. The Ministry of Civil Affairs provided an immediate bulletin and requested that the social organizations and volunteers be subject to the overall command and arrangement of the earthquake relief commanding agency in the disaster area(Ministry of Civil Affairs of the People's Republic of China, 2017). The local government of the disaster area constructed a work station especially for the coordination of the social organizations to register arriving volunteers and social organizations and provide them with jobs, control their sizes, and limit their ability to enter the disaster area according to the situation and requirements of the disaster. These endeavours, which helped to organize the resources as a whole, facilitated the sharing of





information, and engendered cooperation among the professional rescue teams and social organization, made the rescue operations more orderly and effective. The National Earthquake Relief Headquarters deployed a civilian helicopter rescue force that was advantageous in the special disaster environment to join the rescue operations, thereby increasing the degree of cooperation between the military and civilian efforts.

5)      New techniques and technologies, including satellite remote sensing, UAV telemetry, big data, and three-dimensional oblique photogrammetry for rapid disaster evaluations and landslide monitoring, played an important role in the earthquake emergency response and rescue missions. The mobile map service Amap opened a rescue lifeline that marked rescue routes for government sectors and social organization and relief material handout stations. Earthquake thermodynamic maps represent the latest techniques used to provide data support for the study and judgement of the situation of the disaster, as they can

perform comprehensive and statistical analyses, determine the population size and provide real-time changes in the disaster situation by using the push services of mainstream applications installed on smartphones.

**Data availability. Parts of the rescue data are publicly available at http://www.xinhuanet.com/. The emergency response data of all ministries and commissions under the State Council are publicly available at their respective official websites.**

**Author contribution. CH defined the scientific scope of the study. XA and QM collected the data and WW wrote the paper. All authors discussed the results and commented on the paper.**

**Competing interests. The authors have no conflicts of interest to declare.**

**Acknowledgements. Thanks for the remote sensing group in Institute of Crustal Dynamics, China Earthquake Administration, provided the maps of satellite remote sensing.**

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

China Seismological Bureau: China Seismological Bureau Published the Seismic Intensity Map of Ms7.0 Jiuzhaigou Valley Earthquake          of          Sichuan          Province          [EB/OL]: http://www.cea.gov.cn/publish/dizhenj/464/478/20170812211337414565961/index.html, last access: 13 September 2017.





China Seismological Bureau: Earthquake Contingency Plan of the China Seismological BureauContingency Plan for Earthquake Emergency of China Seismological Bureau, 2013.

China Volunteer Service Federation: Pray on Earthquake Disaster Area, Volunteers Rescued Reasonably and With Love in Heart [EB/OL]: https://mp.weixin.qq.com/s?__biz=MzA4MjM2MDgzNA%3D%3D&idx=1&mid=2650827698&sn=38d759525a19c190cca19d0dc7e6a9c9, last access: 22 August 2017.

CISAR: National Earthquake Disaster Emergency and Rescue Team Carryout Rescue Operation on Ms7.0 Jiuzhaigou Valley Earthquake of Sichuan Province [EB/OL]: http://www.toutiao.com/a6452652649400500750/, last access: 13 September 2017.

Cui, D.: Xi Jinping Issued an Important Indication on Ms7.0 Jiuzhaigou Valley Earthquake of Sichuan Province and Demanded that Mobilize Forces to Provide Disaster Relief Rapidly and do our Level Best to Salvage Casualties and Minimise Casualties Furthest, Li Keqiang Made Written on Anti-Quake and Relief Work [EB/OL]: http://politics.people.com.cn/n1/2017/0809/c1001-29458041.html, last access: 22 August 2017.

Department Of Civil Affairs Of Sichuan Province: Department of Civil Affairs has Allocated and Transported a Total of 3000 Cotton Quilts, 3000 Units of Cotton ClothesClothing, 2000 Tents, 1000 Folding Beds, 3000 Sleeping Bags to the Disaster Area Totally [EB/OL]: http://scmz.gov.cn/Article/Detail?id=21586, last access: 13 September 2017.

Gao, C.: Order is more Important than Passion in Earthquake Relief Work with Volunteers Constantly Emerging [EB/OL]: https://www.chinanews.com/sh/2017/08-15/8304864.shtml, last access: 13 September 2017.

General Office of the Aba Prefecture Government: Aba Prefectural Contingency Plan for Earthquake Emergency, 2012.

General Office of the Sichuan Provincial Government: Sichuan Provincial Contingency Plan for Earthquake Emergency, 2012.

General Office of the State Council: Naional Earthquake Contingency PlanState Contingency Plan for Earthquake Emergency, 2012.

General Office of the State Council: National Contingency Plan for Public EmergenciesState Overall Contingency Plan for Precipitating Event, 2006.

General Office of the State Council: State Contingency Plan for Natural Disasters' Relief, 2016.

General Office of the State Council: State Contingency Plan for Paroxysmal Geologic Hazards, 2006.

Liu, Z., and Li, Y.: Analyze on Chinese earthquake emergency response mechanism by Yushu earthquake, China Emergency Rescue, 5, 18-21.

Ministry of Civil Affairs of the People's Republic of China: The No. 411 Announcement from Ministry of Civil Affairs about Social Forces Participate in Earthquake Relief Work Orderly of Ms7.0 Jiuzhaigou Valley Earthquake of Sichuan Province, 2017.

Ministry of Transport of the People's Republic of China: Contingency Plan for Highway Traffic Emergency, 2009.

Qian, F.: People's Armed Police Invested Nearly Two Thousand Soldiers in Jiuzhaigou Valley Earthquake Emergency Rescue [EB/OL]: https://news.youth.cn/jsxw/201708/t20170810_10481852.htm, last access: 13 September 2017.



Standing Committee of the National People's Congress: the Law of the People's Republic of China on Protecting Against and Mitigating Earthquake DisastersLaw on Protecting Against and Mitigating Earthquake Disasters of the People's Republic of China, 2008.

The 10th Standing Committee of the National People's Congress: Emergency Response Law of the People's Republic of China, 2007.

Wang, L.: 30 Central Enterprises Participated in Emergency Rescue in 12 Hours After Jiuzhaigou Valley Earthquake of Sichuan Province [EB/OL]: http://www.jianzai.gov.cn//DRpublish/ztzl/0000000000025485.html, last access: 22 August 2017.

Wang, S.: Revelation on emergency rescue of "4.14" Yushu earthquake by field research on disaster area of Yushu earthquake, Tibet Development Forum, 4, 47-49.

Wang, Y.: They Never Stopped Rescuing in 36 Hours After Jiuzhaigou Valley Earthquake [EB/OL]: http://news.xinhuanet.com/politics/2017-08/10/c_129677481.htm, last access: 13 September 2017.





**Tables**

Table 1: Emergency responses of the major governmental departments.

| Departments | Specific Contents of Emergency Response Work |
|---|---|
| National Headquarters for Earthquake Disaster Mitigation, China Earthquake Administration | 1. The grade I emergency response was launched 1 hour after the earthquake, and the response level was adjusted to a grade II response scenario according to the disaster situation 3 hours after the earthquake. An emergency team was dispatched to the emergency sites to assess and evaluate the disaster characteristics, monitor the aftershocks and analyse trends in the seismic data. The team was composed of experts from the National Earthquake Response Support Service, China Institute of Crustal Dynamics, and other local provincial bureaus such as the Seismological Bureaus of Sichuan Province and Gansu Province. The teams brought UAVs (unmanned aerial vehicles) and seismic observation equipment to the earthquake area to monitor related earthquakes, assess the earthquake intensity and investigate and evaluate the disaster areas. The China International Search & Rescue Team was dispatched to the disaster area to conduct personnel search and rescue. The research institute under the Seismological Bureau carried out a series of studies on the occurrence, fault, focal mechanism and seismic intensity of the earthquake to provide technical support for the earthquake relief efforts. <br> 2. As of August 19 (11 days after the earthquake), the State Council Earthquake Relief Headquarters and the China Earthquake Administration terminated the state-level grade II emergency response to the Jiuzhaigou Ms 7.0 earthquake. |
| China National Committee for Disaster Reduction, Ministry of Civil Affairs of the People's Republic of China | 1. The national grade III disaster relief emergency response was initiated(General Office of the State Council,), and the local Department of Civil Affairs immediately established a working group in the disaster area to assess the disaster characteristics, provide relief need, raise and schedule the delivering of relief supplies, coordinate the social organizations and volunteer teams, receive donations and supervise the relief funds and materials for the disaster areas. <br> 2. The China National Committee for Disaster Reduction, Ministry of Civil Affairs of the People's Republic of China, quickly began to use emergency surveillance UAVs to acquire various types of high-resolution aerial remote sensing images for major natural disasters both at home and abroad and to synergistically monitor and assess the damage to houses, roads and other losses in disaster areas from secondary geological disasters. <br> 3. Within 14 hours after the earthquake, the Ministry of Finance and the Ministry of Civil Affairs allocated a total of 100 million Yuan from the central government's Natural Disaster Subsistence Allowance resettle people in the disaster areas, assist with living needs during the transition period, restore and reconstruct damaged houses and provide solatium for the victims to support their basic livelihood. |





| Departments | Specific Contents of Emergency Response Work |
|---|---|
| National Development and Reform Commission | 1. The Provincial Development and Reform Commission of the disaster area rapidly launched an emergency response, arranged and cooperated with relevant departments to ensure the power supply in the disaster areas, protected the food supplies for the masses, and provided earthquake relief troops and rescue personnel to ensure the stability of prices in the affected areas.<br>2. The Provincial Development and Reform Commission of the disaster area deployed power emergency teams to provide power maintenance for the first time, after which the power supply was restored in Jiuzhaigou County and the surrounding scenic areas 2 hours after the earthquake. As of 07:00 on August 9, 2017 (10 hours after the earthquake), a total of 181 power emergency workers, 50 recovery vehicles, three emergency power-generating vehicles, 4 generators, 10 sets of lighting lamps and lanterns, 4 UAVs, 2 portable satellites for communication and other emergency supplies were invested into repairing the power.<br>3. The 11 gas stations of PetroChina provided more than 500 tonnes of oil products that were used for at least two days in the earthquake area and opened a rescue green channel with conditional stations that started to generate power. All of the Sinopec gas stations along the road to Jiuzhaigou were fully opened with green channels with priority for the earthquake relief vehicles. China Aviation Oil set up emergency teams to arrange rescue personnel and vehicles to provide support for disaster relief flights.<br>4. On August 9, 2017, the National Development and Reform Commission allocated 60 million Yuan (a disaster relief emergency subsidy from the Central Budget Investment) for the reconstruction of the infrastructure and public welfare facilities. |
| Ministry of Transport of the People's Republic of China | 1. The Ministry of Transport of the People's Republic of China immediately initiated a grade II emergency response and deployed emergency personnel from the Department of Transportation of Sichuan Province to unblock roads connecting the airport with the earthquake region so that rescue persons and supplies could be delivered(Ministry of Transport of the People's Republic of China, 2009).<br>2. Key roads and vehicles in and out of the affected areas were controlled. Important traffic points and key crossings were monitored through satellite communications and other technical means to ensure that the rescue crews, relief materials, the wounded and tourists were transported and evacuated in a timely manner.<br>3. The Civil Aviation Administration under the Ministry of Transport immediately initiated a response by implementing and ensuring the normal operations of landing airports for disaster relief personnel and materials as well as for the control and execution of rescue-priority entry and exit flights to ensure that passenger evacuations, rescue crews and supplies for transport were conducted in a timely manner. China Eastern Airlines offered full refunds and free change services for passengers who |





| Departments | Specific Contents of Emergency Response Work |
|---|---|
| | had already purchased a flight to the disaster area. Air China Southwest Branch and China Southern Airlines remained closely aware of the earthquake situation, airports and passenger dynamics to ensure the implementation of relief missions at any time.<br>4. As of 17:30 on August 9, 2017 (20 hours after the earthquake), the state roads and provincial roads in the affected areas were all open. More than 40000 stranded passengers were evacuated, of which nearly 9000 tourists were safely evacuated by emergency vehicles. |
| Ministry of Industry and Information Technology of the People's Republic of China | 1. The Ministry of Industry and Information Technology deployed communications authorities and telecommunications enterprises in the disaster areas to provide emergency communications support. On-site emergency communications and security teams were quickly established to provide communication services for the disaster areas using cable, wireless, and Internet communications pathways. They provided power supply protection for stations that lacked power and service and repaired damaged fibre optic cables as soon as possible to keep satellite telephone communications unblocked and support scheduling and command efforts for first-line rescue teams.<br>2. As of 1:00 on August 9, 2017 (4 hours after the earthquake), the communications to Jiuzhaigou Entrance in addition to 28 base stations and 11 power stations were restored.<br>3. China Mobile, China Unicom and China Telecom immediately launched a tracing hotline to provide satellite communications for disaster relief. In addition, Sichuan users were given compulsory start-up services and were exempted from closing-down services. In addition, free service points were set up in the earthquake areas to provide on-site consultations, free charging, and telephones to receive safety and disaster-warning SMS messages. China Satcom, which is owned by China Aerospace Science and Technology Corporation, also actively mobilized resources to ensure the smoothness of communications and news coverage in the affected areas.<br>4. As of 12:00 on August 10, 2017 (39 hours after the earthquake), a total of 728 security personnel, 168 vehicles and 364 oil machines had been sent by the communications industry. A total of 127 base stations had recovered, but 151 base stations were still not restored (28% of the total number of base stations in Jiuzhaigou County). The local earthquake relief headquarters, local governmental agencies and other departments were provided with 63 satellite phones, 13 emergency circuits and 50 emergency expansion base stations with on-site temporary base stations and satellite equipment to provide communications for command efforts. As a result, the overall operation of the communications network in the disaster area was smooth, and the on-site dispatch and command communications were unimpeded.<br>5. As of 23:00 on August 10, 2017 (50 hours after the earthquake), communications resumed normally in the disaster areas, and the reconstruction of the communications systems officially launched on |





| Departments | Specific Contents of Emergency Response Work |
|---|---|
| | August 16, 2017. |
| National Administration of Surveying, Mapping and Geoinformation | 1. The Chinese Academy of Surveying and Mapping was immediately deployed to provide remote sensing images of the Jiuzhaigou earthquake-affected areas and other emergency support services. The Provincial Administration of Surveying, Mapping and Geoinformation in the disaster area immediately deployed working teams to acquire remote sensing images of the earthquake-affected areas. <br> 2. As of 23:30 on August 8, 2017 (2 hours after the earthquake), an earthquake relief command thematic map and a pre-earthquake image map were completed. In addition, green channels were opened to provide timely surveying and mapping products. <br> 3. On August 9, 2017, the National Administration of Surveying, Mapping and Geoinformation sent professional and technical personnel to the disaster areas with 5 UAVs in addition to communications and measuring equipment to provide high-resolution UAV remote sensing maps for the disaster rescue teams. |
| National Health and Family Planning Commission of the people's Republic of China | 1. Personnel were deployed to immediately understand the disaster situation, guide the localities to conduct rescue operations, provide medical services, arrange national health and medical emergency teams, prevent epidemics and provide psychological experts in the regions surrounding the disaster areas to prepare for other earthquakes. <br> 2. The West China Hospital of Sichuan University sent 8 experts, 2 ambulances and supplies to the disaster areas in Jiuzhaigou 2.5 hours after the earthquake. <br> 3. The hospitals in nearby cities and counties were organized to receive and treat the wounded immediately after the earthquake. In addition, medical teams were dispatched to the disaster area to triage and treat the injured. More than 10 medical teams and nearly 500 medical personnel were deployed to treat the wounded and prevent diseases after the earthquake. |
| Ministry of Land and Resources | 1. A grade I geologic hazard emergency response was immediately initiated(General Office of the State Council, 2006), after which provincial-, state- and prefectural-level working teams were established. Twelve units, 237 specialists and professors and 164 technicians carrying hundreds of sets of equipment were urgently sent to the disaster area to examine landslides and other geological secondary disasters and assess the safeties of temporary shelters. Safety assessments of potential geological hazards were conducted for 339 temporary victim shelters to guide the secure selection of sites and ensure the safety of the masses. <br> 2. Potential geological hazards in the post-earthquake disaster area were examined, and 1258 various potential geological hazards were completely investigated and verified. In addition, remote sensing monitoring and other monitoring technologies were used to assess potential geological hazards to support the safety of rescue and relief efforts. |



| Departments | Specific Contents of Emergency Response Work |
|---|---|
| Ministry of Housing and Urban-Rural Development | 1. The response was rapidly initiated, and specialists and multidisciplinary union working teams were immediately dispatched to the disaster areas to guide earthquake relief efforts.<br>2. Teams were deployed to connect with urban construction bureaus in the disaster areas, verify earthquake casualties, and confirm damage estimates for rural houses, municipal infrastructures and scenic areas. Large rescue machinery and equipment and specialists from major industries were transferred and assembled to construct rescue and relief teams and emergency evaluation teams, and staff support were provided to evaluate emergency losses and emergency restoration efforts for urban and countryside housing and public utilities in the disaster areas.<br>3. The gas pipeline network throughout the disaster areas were examined and repaired to ensure the security and timeliness of the fuel gas supply.<br>4. The post-disaster reconstruction stage began on August 18. A total of 305 temporary shelters were established and for more than 27 thousand people. |
| State Administration of Quality and Technical Supervision | 1. A 96-person rescue team consisting of the Sichuan Province Safety Supervision Bureau and the Sichuan Coal Supervision Bureau was deployed with search and rescue equipment to the disaster areas for rescue operations.<br>2. At 05:40 on August 9, 2017 (8 hours after the earthquake), the rescue team arrived at the field headquarters and began the rescue mission.<br>3. In total, more than 600 personnel from 12 mine emergency rescue teams and 2 hazardous chemical emergency rescue teams of Sichuan Province and more than 400 personnel from 4 mine rescue teams and 2 hazardous chemical emergency rescue teams of the city of Chongqing were conducting and reinforcing Sichuan earthquake relief efforts. |
| China Meteorological Administration | 1. A grade III earthquake disaster meteorological service emergency response was initiated(China Meteorological Administration, 2010), and the National Meteorological Center and Sichuan Province Observatory held a video conference for consultation. Close attention was paid to the latest dynamic weather conditions in the disaster areas and the possibly affected surrounding areas, and meteorological support service was provided for the earthquake relief efforts. At 22:25 (1 hour after the earthquake), the first earthquake disaster relief weather forecast was transmitted from the Jiuzhaigou County Weather Bureau. |
| Ministry of Water Resources | 1. Notice was immediately provided to coherent units in the disaster areas to identify and mitigate potential risks in the earthquake region, monitor changes in the atmosphere, rain conditions, water regime, and torrential mountain disasters, examine the damage situation of reservoir dams at the epicentre and surrounding regions, and conduct preparatory efforts for rescue and relief.<br>2. The Sichuan Province Hydrographic Office immediately established an emergency rush repair headquarters to monitor hydrological emergencies, analyse changes in rivers, determine whether barrier lakes were formed, and provide timely and accurate hydrological information |





| Departments | Specific Contents of Emergency Response Work |
|---|---|
| | for earthquake relief work. |
| | 3. On the third day after the earthquake, security checks were conducted again for major reservoir dams near the disaster areas, and technical support was provided for emergency disposal. The Water Conservancy Department of the disaster area organized specialists to conduct on-site surveys in the Jiuzhaigou scenic areas to discuss the causes of damage and potential solutions and to provide hydrotechnics for the restructuring of the scenic areas. |



**Figure captions**

**Figure 1: Seismic intensity map of the Ms 7.0 earthquake that occurred in Jiuzhaigou County, Sichuan province.**

**Figure 2: Landslides and rockfalls in the disaster area.**

**Figure 3: Remote sensing images acquired by a UAV in the Jiuzhaigou scenic area.**

5 **Figure 4: Comparison of remote sensing images surveying the road damage before and after the earthquake along the No. 301 section of the Dart Road, Jiuzhaigou County, Sichuan province.**





**Figures**

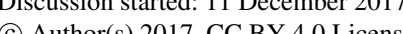

Figure 1




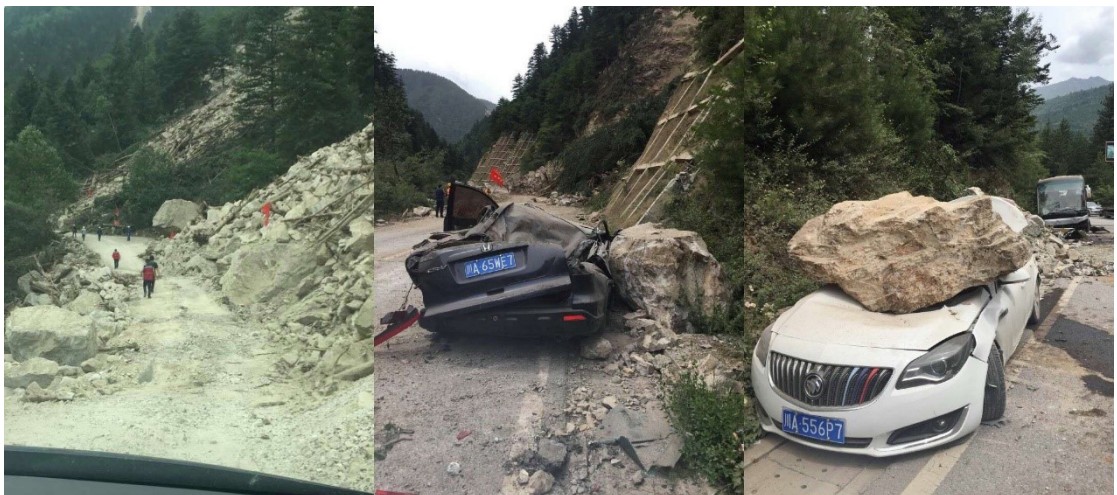

**Figure 2**





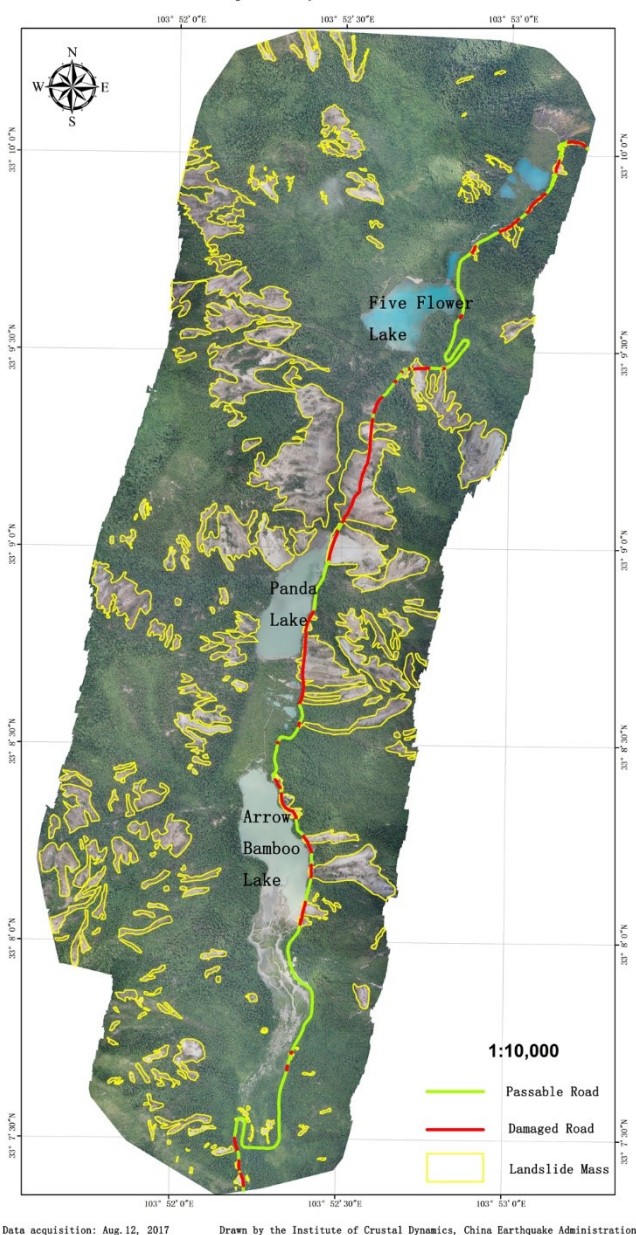

**Figure 3**



Comparison of pre- and post-earthquake remote sensing images of the damaged No. 301
provincial highway within the affected disaster region of Jiuzhaigou County, Sichuan

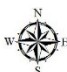

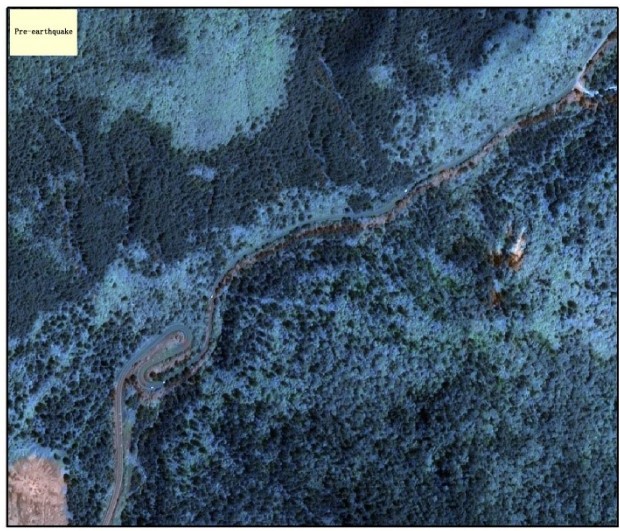

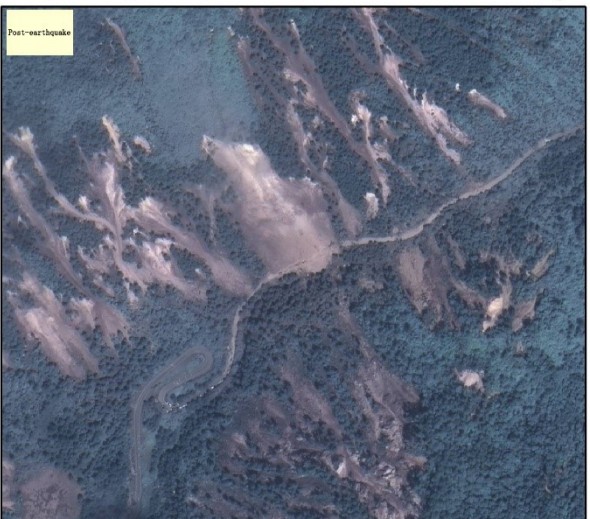

Data source: Satellite Gaofen-2

Data acquisition: Aug.9, 2017

Scale : 1:10,000

Drawn by the Institute of Crustal Dynamics,
China Earthquake Administration

**Figure 4**