# Peer review of "Analysis of the disaster characteristics and emergency response of the Jiuzhaigou earthquake"

_Natural Hazards and Earth System Sciences, 2017_

## Referee Comment (RC1) · Anonymous Referee #1 · 15 Jan 2018

1. General comments This paper provides useful information and implementation of national laws, plans for disaster emergency response in China. Specifically, it describes emergency response activities of all levels of the government, various departments, rescue teams, enterprises and public institutions as well as social organizations in Jiuzhaigou Earthquake area. The paper also discusses lessons and advances in emergency response operation in China in the past decade, and could be referenced by other earthquake-prone countries. The paper is well organized and written, and experiences of this earthquake emergency response is a good example in natural disaster management. 2. Specific comments 2.1 Page 1 line 25: The sentence,"Multiple public health crises have affected China since 2003, including the SARS (severe acute respiratory syndrome) and bird flu epidemics, which highlight the

importance of emergency response teams and activities", could be deleted, because it seems farfetched with earthquake . 2.2 It is right that landslides and rockfalls are main cause for the casualty. It would be nice if the authors give some quantitative data about the number, affected area of landslides. 3. Technical corrections 3.1 Page 3 line 25:"there were approximately 38799 tourists at the nationally famous Jiuzhaigou scenic resort area when the earthquake struck". The number of tourists seems quite accurate, the word"approximately"could be omitted. 3.2 Typing errors, grammar suggestions, etc. are highlighted on the MS.

Please also note the supplement to this comment:
https://www.nat-hazards-earth-syst-sci-discuss.net/nhess-2017-418/nhess-2017-418-RC1-supplement.pdf

**Supplement:**

[Figure]

**Analysis of disaster characteristics and emergency response of  Jiuzhaigou earthquake**

WANG Wei[1], CHEN Hong[2,**], XU Aihui[2], QU Minhao[3]

[1]Institute of Engineering Mechanics, China Earthquake Administration, Harbin 150080, China
5   [2]Institute of Crustal Dynamics, China Earthquake Administration, Beijing 100085, China
[3]National Earthquake Response Support Service, Beijing 100049, China

*Correspondence to*: Chen Hong (chenhongicd@163.com)

**Abstract.** This paper provides a detailed introduction to the disaster situation of the August 8, 2017, Ms 7.0 earthquake that occurred in Jiuzhaigou County, Sichuan Province, China, and specifically describes the emergency response activities of all

10   levels of the government, various departments, rescue teams, enterprises and public institutions as well as social organizations. The characteristics of the earthquake disaster and the emergency responses are analysed and summarized. The emergency response activities during  Jiuzhaigou earthquake exhibited three distinct characteristics relative to those during similar earthquakes. The response level and resource mobilization were appropriately and sensibly adjusted according to the development of the disaster, and various departments worked closely together to conduct multi-sector joint rescue efforts.

15   Moreover, professional rescue forces and participating social organizations were more rationally mobilized. A set of effective disaster relief command and coordination mechanisms were established for cooperation between multiple departments and the participation of many social organizations under the leadership of the local government. Finally, new and more effective technologies played an important role in the emergency response and rescue efforts following this earthquake.

**Key words**: Jiuzhaigou earthquake; earthquake disaster; emergency response; emergency rescue

20   **Copyright statement: The author's copyright for this publication is transferred to Institute of Engineering Mechanics, China Earthquake Administration.**

[revised manuscript text omitted]

---

## Referee Comment (RC2) · D. E. Alexander (Referee) · 16 Jan 2018

Page 1, Line 15: "... were more rationally mobilized" - more than what?

Section 1: I am wondering whether China uses an all-hazards approach to emergency preparedness and response or whether it has specific earthquake emergency plans. I also wonder whether a diagram, such as a flow-chart, might help clarify the structure of emergency management in China, including the lines of command and control.

Pge 3, Line 26: "...when the earthquake struck month." - delete 'month'

Line 27: "rolling stones" - rockfalls (or possible debris avalanches)

Page 4, lines 5-6: "...a seismic fortification level of 8 degrees," - it is not clear what this

means. Seismic resistance, evidently, but eight degrees of what?

Page 5, lines 14-15 (and line 25): "...an "8.8" Jiuzhaigou earthquake relief emergency headquarter[s]" - what does this mean?

Pages 7-8: "...at dusk, did provided orderly maintenance" - substitute 'they' for 'did'

Section 3 resembles many earthquake situation reports I have read over the years. Very little in it is novel. The authors describe a typical post-earthquake convergence reaction.

Page 9, lines 17-19: "By learning from the experiences of the emergency response activities during the Ms 7.1 Yushu earthquake in 2010 and the Ms 7.0 Lushan earthquake in 2013, the Chinese earthquake catastrophe emergency response system has become more mature, and the response mechanisms have greatly improved." - This is important and could provide the basis of an interesting and penetrating analysis.

This paper is nicely written and gives a careful description of the event that the authors report on. However, it seriously lacks depth. In the literature there are various analyses of emergency organisation and emergency response and there is a theoretical side to this. There is no reference to any of this work anywhere in this paper, which is almost entirely a description of an event and the response to it. Various works by Ron Perry, Michael Lindell, David Alexander, David Neal, David Godschalk, William Wallace and others is relevant here.

Readers will be intensely interested to learn about the organisation of emergency response in China (there are few up-to-date sources in English on this) and also about how it has evolved since the signal event of 2008. However, in this manuscript the critical analysis is muted or lacking and the comparison with events, or better still systems, elsewhere is missing.

There are various theoretical aspects that are of interest, as well: the tension between devolution and centrism, incident command versus the hierarchical model, command

versus collaboration, the command function principle versus the support function principle, and so on. More depth is seriously needed, and it could lead to a very interesting and worthwhile paper.

I would be happy to help the authors with references if they have trouble obtaining the literature.

---

## Author Comment (AC1) · 22 Jan 2018

Dear Sir or Madam, Thank you for your comments for this paper. Your suggestions will help us to improve our research paper. Just like what you said, our purpose was to summarize lessons and advances in emergency response operation by this Jiuzhaigou Earthquake, and then analyze the changes and advances in all phases of earthquake emergency management in China in the past decade since Wenchuan Earthquake. It is highly important in further research on earthquake emergency management in China and is a good example in natural disaster management in China and could be referenced by other earthquake-prone countries. To the detail questions, our answers are as follows: 2.1 The reason we used SARS was that this event started the modern emergency management in China. A comprehensive emergency management system

(Contingency Plan System, Legal System, Organization System, Operation Mecha-nism) was formed gradually after 2003. As earthquake is the worst disaster in China, China formed emergency response system specially for earthquakes, which including the Earthquake Contingency Plan System, the Earthquake Emergency Command Sys-tem, etc. 2.2 I will try to find out whether it has official accurate quantitative data about the number, affected area of landslides. At present, the data collected in my hands are approximately 1880 landslides and approximately $15\times104m2$ areas by remote sens-ing data analysis from our research institutions. 2.3 I will take your advice. In the end, thank you for your help again, and this paper will benefit greatly from your thoughtful reviews.

---

## Author Comment (AC2) · 22 Jan 2018

Dear Sir, Thank you for your comments for this paper. Your suggestions will help us to improve our research paper, but there are some issues that require further discussion. The original intention we wrote this paper was to summarize lessons and advances in emergency response operation by this Jiuzhaigou Earthquake, and then analyze the changes and advances in all phases of earthquake emergency management in China in the past decade since Wenchuan Earthquake. For this, we provided useful information and implementation of national laws, plans for disaster emergency response in China, and specifically described emergency response activities of all levels of the government, various departments, rescue teams, enterprises and public institutions as well as social organizations during Jiuzhaigou Earthquake response. The response

activities exhibited three distinct characteristics relative to those during similar earthquakes. It was a successful emergency response event and reflected the changes and advances in all phases of earthquake emergency management in China after the Wenchuan Earthquake, which we discussed in the conclusion of the paper. It is highly important in further research on earthquake emergency management in China and is a good example in natural disaster management in China and could be referenced by other earthquake-prone countries. To the detail questions, our answers are as follows: 1. More rational than the past earthquake emergency response. 2. China uses departmental system of management to deal with different disasters. Chinese contingency plan system covers specialized emergency plans of all government sectors (the Ministry of Civil Affairs, the Ministry of Finance, the Ministry of Land and Resources, the Ministry of Housing and Urban Construction, the Ministry of Transport, the National Health and Family Planning Commission, China Earthquake Administration etc.) and all government levels (national, provincial, municipal, prefectural and community level). The contents include disaster monitoring and early warning, prevention and preparation, emergency handling, disaster relief, and rehabilitation and reconstruction. Detailed measures and working regulations are worked out by the relevant government departments in line with the specialized plans and their respective responsibilities. In the wake of a major natural disaster, under the unified leadership of the State Council, the relevant departments with different focuses shall act in coordination and launch emergency response plans to guide disaster control and relief work. The governments of the affected areas shall immediately start emergency response measures and set up a local disaster emergency response command with the heads of the local governments serving as the chief commanders, and leaders of relevant departments as members, to jointly draw up emergency plans and measures, organize field emergency response work, and report disaster details and work progress to governments of higher levels and relevant departments. In the case of earthquakes, China has the National Earthquake Contingency Plan and the Earthquake Contingency Plan of the China Earthquake Administration, then all levels of the government of China,

relevant departments, departments in charge of earthquake-related work, organizations directly under the China Earthquake Administration, key enterprises and institutions, schools, hospitals, communities and condensed populated places have established their own earthquake contingency plans. China has large population density and some of the most serious earthquake damage statistics and casualty rates in the world. The earthquakes that occur throughout China are widespread, generally have larger magnitudes with shallow focus depths, and are typically characterized by substantial hazards. Earthquakes with M$\geq$5 have occurred in every province of China over the course of recorded history. So the Chinese government places great importance on the prevention of earthquakes and the mitigation of their associated hazards. And established the State Earthquake Control and Rescue Head-quarters as the high authority responsible for the earthquake work, which consists of competent department (the China Earthquake Administration) and other ministries and commissions (the Ministry of Civil Affairs, the Development and Reform Commission, the Ministry of Finance, the Ministry of Land and Resources, the Ministry of Housing and Urban Construction, the Ministry of Transport, and the National Health and Family Planning Commission, etc.) and the office of the head-quarters is set up in the China Earthquake Administration. When the earthquake occurred, launched different response level according to different earthquake disaster levels. The Earthquake Administration as the competent department organized and coordinated the emergency rescue operation. We drew a flow chart to show the lines of command and control of the Earthquake Emergency Command System. Fig.1 Earthquake Emergency Command System 3. I agree, delete "month". 4. "Rockfalls", I will take your advice. 5. 8 degree of earthquake fortification (seismic precautionary intensity). 6. It is the name of the command center. 7. I agree, substitute "they" for "did". Finally, thank you for your help again. And thank you for this "interactive comment", which let me learn more about what kind of up-to-date sources in English about China you may be interested in. And I wish to keep contact with you about research progress in earthquake emergency management.

[Figure]

```
                    ┌─────────────────────────┐
                    │    the State Council    │
                    └─────────────────────────┘
             ┌──────────┐          ┌──────────┐
             │ Suggest  │          │  Launch  │
             │ Launch   │          └──────────┘
             └──────────┘
```

| Extraordinarily serious earthquake disaster | State Earthquake Control and Rescue Head-quarters Leader from the State Council takes the chair | Level I emergency response | Make earthquake emergency response decision / Organize head-quarters member units to carry out emergency rescue |
| Serious earthquake disaster | the director general of China Earthquake Administration takes the chair | Level II emergency response | Offer emergency proposal to the State Council / Entrusted by the State Council Coordinate units concerned to carry out emergency rescue |
| Major earthquake disaster | the deputy director general of China Earthquake Administration takes the chair | Level III emergency response | Offer emergency proposal to the State Council / Coordinate working teams dispatched by units concerned on the site |
| General earthquake disaster | the deputy director general of China Earthquake Administration takes the chair | Level IV emergency response | Offer earthquake trend estimation / Dispatch earthquake on-site emergency task force |

(China Earthquake Administration)

**Fig. 1.** Earthquake Emergency Command System

---

## Author Response (AR1)

**Reply to Author Queries**

Journal: Natural Hazards and Earth System Sciences

Manuscript: nhess-2017-418

**1. Reply to Anonymous Referee #1**

Q1 Page 1 line 25: The sentence, "Multiple public health crises have affected China since 2003, including the SARS (severe acute respiratory syndrome) and bird flu epidemics, which highlight the importance of emergency response teams and activities", could be deleted, because it seems farfetched with earthquake.

A1 The reason we used SARS was that this event started the modern emergency management in China. A comprehensive emergency management system ("One Planning Plus Three Systems" - emergency response plan, emergency legislation system, emergency institutional system and the emergency regulatory system.) was formed gradually after 2003.

Q2 It is right that landslides and rockfalls are main cause for the casualty. It would be nice if the authors give some quantitative data about the number, affected area of landslides.

A2 OK. I have added "The earthquake triggered at least 622 co-seismic landslides within an image cover 3919km$^2$" to page 3 line 18.

Q3 Page 3 line 25:"there were approximately 38799 tourists at the nationally famous Jiuzhaigou scenic resort area when the earthquake struck". The number of tourists seems quite accurate, the word "approximately" could be omitted.

A3 OK. I substitute "approximately 40 thousand tourists" for it.

Q4 Typing errors, grammar suggestions, etc. are highlighted on the MS.

A4 OK.

**2. Reply to D. E. Alexander (Referee)**

Q1 Page 1, Line 15: "... were more rationally mobilized" - more than what?

A1 Professional rescue forces and participating social organizations were more rationally mobilized than during past earthquake emergency responses.

Q2 Section 1: I am wondering whether China uses an all-hazards approach to emergency preparedness and response or whether it has specific earthquake emergency plans. I also wonder whether a diagram, such as a flow-chart, might help clarify the structure of emergency management in China, including the lines of command and control.

A2 I rewrite the introduction to explain the China's emergency management system. Its approach can be summarized as "One Planning Plus Three Systems" - emergency response plan, emergency legislation system, emergency institutional system and the emergency regulatory system. And explain the specific earthquake emergency plans. At the beginning of the chapter 3, I explain the China's

earthquake emergency response command and control system and give a flow-chart in Fig.5.

Q3 Page 3, Line 26: "...when the earthquake struck month." - delete 'month'
A3 OK.

Q4 Line 27: "rolling stones" - rockfalls (or possible debris avalanches)
A4 rockfalls.

Q5 Page 4, lines 5-6: "...a seismic fortification level of 8 degrees," - it is not clear what this means. Seismic resistance, evidently, but eight degrees of what?
A5 These buildings were constructed with a seismic precautionary intensity of 8-degrees, and thus possessed high levels of seismic fortification and favourable shock resistance.

Q6 Page 5, lines 14-15 (and line 25): "...an "8.8" Jiuzhaigou earthquake relief emergency headquarter[s]" - what does this mean?
A6 It is the name of provincial Earthquake Control and Rescue Headquarters in the disaster area, and I substitute "established a provincial ECRH called the 8.8 ECRH" (page 5 line 1) for it.

Q7 Pages 7-8: "...at dusk, did provided orderly maintenance" - substitute 'they' for 'did'
A7 OK.

Q8 Page 9, lines 17-19: "By learning from the experiences of the emergency response activities during the Ms 7.1 Yushu earthquake in 2010 and the Ms 7.0 Lushan earthquake in 2013, the Chinese earthquake catastrophe emergency response system has become more mature, and the response mechanisms have greatly improved." - This is important and could provide the basis of an interesting and penetrating analysis.
A8 I have added a chapter 3.6 "Comparison of the emergency response to four large earthquakes" and Table 2 to analyze the improvement of the earthquake emergency response.

Q9 Readers will be intensely interested to learn about the organisation of emergency response in China (there are few up-to-date sources in English on this) and also about how it has evolved since the signal event of 2008.
A9 Thank you for your help, this paper will benefit greatly from your thoughtful reviews. And I supplemented relevant content in chapter 1, the beginning of chapter 3, chapter 3.6, Table 2, and Fig.5. In addition to this, I also revised other content, and I hope to get your professional advice.

**Analysis of the disaster characteristics and emergency response of the Jiuzhaigou earthquake**

WANG Wei[1], CHEN Hong[2**], XU Aihui[2], QU Minhao[3]

[1]Institute of Engineering Mechanics, China Earthquake Administration, Harbin 150080, China
5 [2]Institute of Crustal Dynamics, China Earthquake Administration, Beijing 100085, China
[3]National Earthquake Response Support Service, Beijing 100049, China

*Correspondence to*: Chen Hong (chenhongicd@163.com)

**Abstract.** China's earthquake emergency response system has been improved by lessons learned from multiple earthquakes. This paper focuses on the $M_s$ 7.0 earthquake that occurred in Jiuzhaigou County, Sichuan Province, China, on 8 August 2017 10 and assesses the emergency response activities of all levels of government as well as various departments, rescue teams, enterprises and public institutions and social organizations. The emergency response is compared to other large earthquakes that occurred in China in recent years. The lessons learned from these experiences can inform the emergency response to future disasters. The characteristics of the Jiuzhaigou earthquake and the emergency responses after the event are analysed. The response level and resource mobilization were appropriately adjusted as the disaster developed, and various departments 15 worked together to conduct multi-sector joint rescue efforts. Additionally, professional rescue forces and participating social organizations were more rationally mobilized than during past earthquake emergency responses. A set of effective disaster relief command and coordination mechanisms were established to facilitate cooperation between multiple departments and social organizations under the leadership of the local government. Finally, new and more effective technologies played an important role in the emergency response and rescue efforts following the earthquake.

20 **Copyright statement:**

**1 Introduction**

Since the early 1980s, more than 30 laws and regulations on disaster prevention and reduction have been promulgated regarding earthquake, meteorological, and flood disasters, water pollution, soil desertification, forest fires, and environment protection. However, China's modern emergency management was preliminary formed during the SARS (severe acute respiratory 25 syndrome) event since 2003, which affected as many as 26 provinces and 5327 persons with death rate of 6.53% (Li et al., 2004), highlighting the importance of emergency response. Advanced emergency management concept on all-hazards approach, integrated emergency management system, emergency lifecycle, and emergency planning (Coombs, 1999; Alexander, 2002; Perry and Lindell, 2003; S. Greiving et al., 2012;Alexander, 2015), were used for reference by China. Then, China's emergency management system has undergone a significant change, which can lead to an integrated response during

**Copyright statement: The author's copyright for this publication is transferred to Institute of Engineering Mechanics, China Earthquake Administration. .**

[revised manuscript text omitted]

**3.5 Emergency response of social organizations**

The Red Cross Society of China, the Sichuan Red Cross Society, and the Chinese Red Cross Foundation all launched emergency responses when the earthquake occurred. They quickly allocated relief goods and emergency funds in support of the emergency rescue efforts.

Less than 2 hours after the earthquake, the first social organization rescue team had already arrived in the disaster area and immediately began rescue efforts. As of 12 August, a total of 219 social organizations and more than 2288 volunteers had registered at the 8.8 SC (Gao, 2017). Social organization rescue operations acted in accordance with the unified arrangement of the 8.8 SC, and consequently, they entered and evacuated the disaster area on time. Social organization rescue teams identified hidden dangers, transferred disaster-affected people, assisted with the evacuation of visitors, and provided orderly maintenance and psychological counselling for stranded visitors in temporary shelters. The 8.8 SC coordinated the efforts of all involved parties, constructed volunteer service sites near vital communications lines, stations, hospitals and other crucial locations, and dispatched more than 1200 volunteers to provide volunteer service in the towns, including transporting and distributing relief materials, guiding traffic, providing psychological comfort, relocating stranded visitors, and caring for the wounded (China Volunteer Service Federation, 2017).

Among these teams, the Blue Sky Rescue Team sent a total of 12 teams and 259 team members, performed search and rescue operations, evacuated people, constructed tents, offered medical assistance, verified information, distributed relief materials, transferred victims, and generated disaster situation assessment reports. They successfully discovered 10 victims and the remains of 5 others who had been trapped in the scenic areas without communication for more than 40 hours. All the teams in the Blue Sky Rescue Team evacuated from Jiuzhaigou before 20:00 on 12 August according to the 8.8 ECRH. Helicopter rescue forces also participated in the Jiuzhaigou earthquake rescue operations. Due to physical restrictions on land-sky military helicopters, the SECRH dispatched two airbus helicopters from Xilinfengteng General Aviation Co. Ltd. to the dead zone of the earthquake disaster area to conduct search and rescue operations for trapped, sick and wounded people, during which a total of 30 victims were saved.

**3.6 Comparison of the emergency response to four large earthquakes**

Table 2 compares of the disaster situation (Zheng et al., 2010; Zheng et al., 2011; Chen et al., 2014; Zheng and Zheng, 2015) and emergency response data (Yang and Chen, 2008; Cheng et al., 2010; Deng, 2010; Yang, 2010; Wang and Chen, 2012; Editorial Office, 2013; Lu and Xu, 2014; Yang et al., 2014) for four major earthquakes. The data in table 2 shows that each emergency response improved relative to the previous ones. During this decade, the time of earthquake rapid reporting decreased from 19 minutes to 25 seconds, and the time of the first press conference decreased from 26.5 hours after an earthquake to 2 hours. Thus, disaster relief information could be released in a timely manner, helping to guide public opinion and thus maintain social stability in a disaster area. While emergency surveying and mapping improved from a technological perspective. The time of disaster assessment results release decreased from 109d to 4d. The recovery of lifeline is faster with

[revised manuscript text omitted]


**Team list**

**Author contributions**

CH defined the scientific scope of the study. XA and QM collected the data, and WW wrote the paper. All authors discussed the results and commented on the paper.

5    **Competing interests**

The authors have no conflicts of interest to disclose.

**Disclaimer**

**Acknowledgements**

We thank the remote sensing group in the Institute of Crustal Dynamics, China Earthquake Administration, for providing the

10   satellite remote sensing maps.

China Volunteer Service Federation: Pray for Earthquake Disaster Area, Volunteers Rescued Reasonably and With Love in Heart [EB/OL]: https://mp.weixin.qq.com/s?__biz=MzA4MjM2MDgzNA%3D%3D&idx=1&mid=2650827698&sn=38d759525a19c190cca19d

**Marginal change-tracking annotations:**


[revised manuscript text omitted]


| Departments | Specific Aspects of Emergency Response Work |
|---|---|
| Ministry of Water Resources | 1. Notice was immediately provided to units in the disaster area to identify and mitigate potential risks in earthquake region, monitor changes in the atmosphere, including rain conditions, water regime, and torrential mountain disasters, examine the damage to reservoir dams at the epicentre and surrounding regions, and conduct preparatory efforts for rescue and relief.
 2. The Sichuan Province Hydrographic Office immediately established an emergency rush repair headquarters to monitor hydrological emergencies, analyze changes in the rivers, determine whether quake lakes had formed, and provide timely and accurate hydrological information for earthquake relief work.
 3. On the third day after the earthquake, security checks were repeated for the major reservoir dams near the disaster area, and technical support was provided for emergency disposal. The Water Conservancy Department of disaster area organized specialists to conduct on-site surveys in the Jiuzhaigou scenic areas to discuss the causes of damage and potential solutions and to provide hydrotechnics for the restructuring of the scenic areas. |

**Table 2: Comparison of disaster situation and emergency response data to four earthquakes.**

| | Wenchuan Earthquake | Yushu Earthquake | Lushan Earthquake | Jiuzhaigou Earthquake |
|---|---|---|---|---|
| Epicentre | 31.021°N 103.367°E Sichuan province | 33.165°N 96.629°E Qinghai province | 30.284°N 102.956°E Sichuan province | 33.193°N 103.855°E Sichuan province |
| Time | 14:28, Monday, 12 May, 2008 | 07:49, Wednesday, 14 April, 2010 | 08:02, Saturday, 20 April, 2013 | 21:19, Tuesday, 8 August, 2017 |
| Magnitude | $M_s$ 8.0/$M_w$ 7.9 | $M_s$ 7.1/$M_w$ 6.9 | $M_s$ 7.0/$M_w$ 6.6 | $M_s$ 7.0/$M_w$ 6.5 |
| Depth | 19 km | 14 km | 13 km | 20 km |
| Fatalities | 69227 | 2968 | 196 | 25 |
| Missing | 17923 | 270 | 21 | 6 |
| Injuries | 375783 | 11000 | 13019 | 525 |
| Direct economic loss | ¥8,523 hundred million | ¥228.47 hundred million | ¥665.14 hundred million | ¥224.5 hundred million |
| Main secondary disaster | Quake lakes | Landslides | Landslides and rockfalls | Landslides and rockfalls |
| Earthquake rapid report | 19m | 15m | 55s | 25s |
| Headquarters | State level in Beijing | Provincial level in disaster area | Provincial level in disaster area | Provincial level in disaster area |
| Emergency surveying and mapping | Non-systematic | 3h | 2h | 2h |
| Disaster assessment result | 109d | 8d | 6d | 4d |
| First press conference | 26.5h | 32h | 3.5h | 2h |
| Rescue team response | 30m | 10m | 5m | 5m |
| Power restoration | 105h | 67h | 28h | 2h |
| Communication restoration | 48h | 25h | 29h | 4h |
| Traffic restoration | 79h | 24h | 33h | 20h |
| Rescue forces | More than 146 thousand | Approximately 15 thousand | More than 18 thousand | Approximately 5 thousand |
| Volunteers | Approximately 1.3 million | More than 20 thousand | Approximately 24 thousand | Approximately 2.3 thousand |
| Reconstruction overall planning | 4 months | 2 months | 2.5 months | 3 months |

**Figure captions**

**Figure 1: Seismic intensity map of the $M_S$ 7.0 earthquake that occurred in Jiuzhaigou County, Sichuan Province.**

**Figure 2: Landslides and rockfalls in the disaster area.**

**Figure 3: Post-seismic remote sensing images acquired by a UAV in the Jiuzhaigou scenic area.**

5 **Figure 4: Comparison of remote sensing images surveying the road damage before and after the earthquake along the No. 301 provincial highway, Jiuzhaigou County, Sichuan Province.**

**Figure 5: China's earthquake emergency response command and control system.**

**Figures**

[Figure]

[Figure]

Figure 1

[Figure]

**Figure 2**

[Figure]

1:10,000

——— Passable Road

——— Damaged Road

▭ Landslide Mass

Data acquisition: Aug.12, 2017    Drawn by the Institute of Crustal Dynamics, China Earthquake Administration

**Figure 3**

[Figure]

[Figure]

[Figure]

Data source: Satellite Gaofen-2
Data acquisition: Aug.9, 2017

Scale : 1:10,000

Drawn by the Institute of Crustal Dynamics,
China Earthquake Administration

**Figure 4**

[Figure]

Figure 5

---

## Referee Report (RR1)

[Figure]

Figure 5

[referee-annotated manuscript omitted]